# Peer review of "Biodegradation Rate of EDTA and IDS and Their Metal Complexes"

_horticulturae, doi:10.3390/horticulturae9060623_

Round 1

Reviewer 1 Report

This paper describes results of the biodegradation rate of EDTA and IDS and their metal complexes. It seems to be well covered, described and presented. 

The symbol L is used for liter as in mg/L, not mg/l. Page 1 line 33, remove and, move it to biodiversity and greenhouse. LIne 36 fertilizers is their not the, Line 38, microelements not microelemets.  Line 53 page 2 too many dots, 62, co-authors found, 65 biomass in an hour, 80, and pyri-dine....line 98 inexpensive not unexpensive Page 3 line 102 used as complex. line 112 remove "and taken....EDTA" as it is not clear. page 7 line 240, environment, line 241, supposed to not influence, line 250 metals.... the rest is just l to L.

.

Author Response

The authors express their gratitude to the Reviewer for a careful reading of the paper, valuable comments, the correction of which will make this paper better and avoid similar errors/inaccuracies of wording in the future.

Reviewer 2 Report

The manuscript presents the results obtained from studies of biodegradation rate, plant growth stimulation  (one plant test: Raphanus sativus) and ecotoxicity towards Daphnia magna and Chlorella vulgaris of iminodisuccinic acid (IDS) to be considered as an alternative for EDTA complexation agent for microelements used in fertilizers formulation.  Results obtained showed that the biodegradation rate of  IDS and microelements complexed with IDS  rate is greater than  EDTA. The germination index for IDS concentration of 100 mg/L (both in pure form and combination with microelements) was higher than those of EDTA. It can be concluded that IDS is a promising agent to chelate microelements used in plant nutrition. Indeed, after manuscript reading, we can conclude that the IDS appear to be a promising agent for chelating microelements used in fertilizers formulations. As an advantage, the use of IDS for microelements complexation has an effect the plant growth stimulation. Additionally, IDS is higher biodegradable than EDTA both as a chemical compound and as a complex that prevents its accumulation in the environment. Besides, the microelements complexed with IDS  possess lower ecotoxicity towards aquatic organisms such as D. magna and Chl. vulgaris.

I consider that the article is well-written and concise, and the methodology used is correct and well-described. The results and discussions are presented concisely. The subject is of great interest to scientists from the horticultural fields. I recommend this article for publication, with minor revisions. In this regards the authors must read with attention the manuscript and  solve the following aspects:

1) The sentences cannot begin with an abbreviation;

2) The names microorganisms, microalgae and planktonic crustaceans used in this study must be written in italics ( row 61; rows 298, 299) 

3) Avoid the use of words like ''solo'';  (row 295) this word must be replaced with something else.

Author Response

(The authors gave the same response as above.)

Reviewer 3 Report

The present Manuscript “Biodegradation rate of EDTA and IDS and their metal complexes” written by Beltyukova et al., is an interesting idea. The manuscript is written well and the authors got very good outcomes. However, some changes are needed before the final decision. Like 53; please delete one full stop.

Line 102: Potentially, those complexes can be used as complex micro-fertilizers in agriculture. Please revisit this sentence.

I suggest the authors to add an attractive objective along with the hypothesis in the revised version.

Figure 1: Authors can add the letters on bars to show the statistical significance among treatment means.

The quality of the figure is very poor, please use the origin/sigma plot or any other software to make good-quality graphs. Line 217-218: It should be noted that no significant differences (p>0.05) in the germination index for EDTA and IDS at a concentration of 1 g/l were found. These lines must be revised.

The discussion section is written well, however, it needs improvement to bring novelty to this section.

In the conclusion section, I suggest the authors to add future research directions.

The manuscript needs moderate English editing.

Author Response

(The authors gave the same response as above.)

Round 2

Reviewer 3 Report

The authors have substantially improved the MS, therefore, it can now be accepted for publication. 

Please add letters on the bars of Figure 3 (Toxicity to (a) - Ch. vulgaris and (b) - D. magna individual chelating agents and complex 241 fertilizers based on them)

The quality of the English Language is fine.